# LOXL2—A New Target in Antifibrogenic Therapy?

**DOI:** 10.3390/ijms20071634

**Published:** 2019-04-02

**Authors:** Angela Puente, Jose Ignacio Fortea, Joaquin Cabezas, Maria Teresa Arias Loste, Paula Iruzubieta, Susana Llerena, Patricia Huelin, Emilio Fábrega, Javier Crespo

**Affiliations:** 1Digestive Disease Department, Marqués de Valdecilla University Hospital, Cantabria University, 39008 Santander, Spain; joseignacio.fortea@scsalud.es (J.I.F.); joaquin.cabezas@scsalud.es (J.Ca.); mteresa.arias@scsalud.es (M.T.A.L.); paula.iruzubieta@scsalud.es (P.I.); susana.llerena@scsalud.es (S.L.); patricia.huelin@scsalud.es (P.H.); emilio.fabrega@scsalud.es (E.F.); javier.crespo@scsalud.es (J.Cr.); 2Health Research Institute Marques de Valdecilla (IDIVAL), 39008 Santander, Spain

**Keywords:** fibrosis, LOXL2, portal hypertension, hepatic stellate cells, regression cirrhosis

## Abstract

The concept of liver fibrosis and cirrhosis being static and therefore irreversible is outdated. Indeed, both human and animal studies have shown that fibrogenesis is a dynamic and potentially reversible process that can be modulated either by stopping its progression and/or by promoting its resolution. Therefore, the study of the molecular mechanisms involved in the pathogenesis of liver fibrosis is critical for the development of future antifibrotic therapies. The fibrogenesis process, common to all forms of liver injury, is characterized by the increased deposition of extracellular matrix components (EMCs), including collagen, proteoglycans, and glycoproteins (laminin and fibronectin 2). These changes in the composition of the extracellular matrix components alter their interaction with cell adhesion molecules, influencing the modulation of cell functions (growth, migration, and gene expression). Hepatic stellate cells and Kupffer cells (liver macrophages) are the key fibrogenic effectors. The antifibrogenic mechanism starts with the activation of Ly6C^high^ macrophages, which can differentiate into macrophages with antifibrogenic action. The research of biochemical changes affecting fibrosis irreversibility has identified lysyl oxidase-like 2 (LOXL2), an enzyme that promotes the network of collagen fibers of the extracellular matrix. LOXL2 inhibition can decrease cell numbers, proliferation, colony formations, and cell growth, and it can induce cell cycle arrest and increase apoptosis. The development of a new humanized IgG4 monoclonal antibody against LOXL2 could open the window of a new antifibrogenic treatment. The current therapeutic target in patients with liver cirrhosis should focus (after the eradication of the causal agent) on the development of new antifibrogenic drugs. The development of these drugs must meet three premises: Patient safety, in non-cirrhotic phases, down-staging or at least stabilization and slowing the progression to cirrhosis must be achieved; whereas in the cirrhotic stage, the objective should be to reduce fibrosis and portal pressure.

## 1. Introduction

Chronic liver disease is a major cause of morbidity and mortality in developed countries. Regardless of the causal agent of the liver disease (chronic viral hepatitis B or C, alcoholic or nonalcoholic steatohepatitis), all types of liver disease have a common pathophysiological mechanism, fibrogenesis, cirrhogenesis, and the subsequent development of portal hypertension syndrome. After acute injury, liver regeneration can be completed in a short time; however, when the noxa persists over time, a chronic wound-healing response is established, leading to the replacement of parenchymal cells by extracellular matrix components (EMCs) [1,2,3]. Although this mechanism could be initially beneficial, progressive accumulation of EMC will gradually generate abnormalities in the vascular architecture and a scarred parenchyma with a long-term outcome of hepatic dysfunction and cirrhosis.

The concept of liver fibrosis and cirrhosis being static and therefore irreversible is outdated. Indeed, both human and animal studies have shown that fibrogenesis is a dynamic and potentially reversible process that can be modulated either by stopping its progression and/or by promoting its resolution [4]. Therefore, the study of the molecular and cellular mechanisms involved in fibrogenesis is critical for establishing future antifibrotic therapies [4,5]. Furthermore, different patterns of fibrosis progression have been described on the basis of their etiology. In chronic viral hepatitis B and C, the presence of interface hepatitis and portal–central vein bridging is characteristic; in alcohol-related liver fibrosis, the deposition of ECM (extracellular matrix) in the space of Disse around sinusoids (capillarization of the sinusoids) or hepatocytes accompanied with perisinusoidal or pericellular fibrosis (also in nonalcoholic fatty liver disease) predominates. Finally, biliary fibrosis is associated with the proliferation of bile ductules and periductular myofibroblasts, which leads to the formation of portal–portal fibrotic septa [5,6,7].

Mediators of liver injury could be different depending on the etiology of the liver disease; for example, in alcoholic liver disease (ASH) we found lipopolysaccharide (LPS)-binding protein or Fe; in viral hepatitis, HCV/HBV proteins; in non alcoholic steatohepatitis (NASH), glucose and adipocytokines; in Primary biliar cholangitis (PBC), bile acids; and in hemochromatosis, excess of iron [5].

## 2. Mechanism of Liver Fibrosis

The liver parenchyma consists of its own epithelial cells (hepatocytes), endothelial cells, and non-parenchymal cells, including hepatic stellate cells (HSCs) and Kupffer cells (KCs) [2]. The hepatic microvascular functional unit is the liver sinusoid. It is formed of an endothelial lining, distinguished by fenestrated pores, and is separated from the hepatocytes by the subendothelial space of Dissé, where HSCs reside [6]. This space allows the metabolic exchange between blood and liver cells. The fibrogenesis process, common to all forms of liver injury, is characterized by the increased deposition of EMCs, including collagen (collagen types I, IIIand IV), proteoglycans, and glycoproteins (laminin and fibronectin 2). These changes in the composition of the EMCs lead to an alteration of their interaction with cell adhesion molecules, influencing the modulation of cell functions (growth, migration, and gene expression) [7]. HSC and Kupffer cells (liver macrophages) are the key elements in fibrogenesis, and the transdifferentiation to Ly6C^high^ macrophages constitutes the basis for the regression of fibrosis. The mechanism of hepatic fibrosis can be divided into three parts (Figure 1).

1. Acute liver damage triggers the activation of Kupffer cells, which coordinate the regenerative response; however, if chronic damage persists, over-activation of these cells results in the release of proinflammatory cytokines (CCL2 and CCL5) and a stimulation of the bone marrow for the generation of activated Ly6C^high^ [8,9].

2. This inflammatory magma induces the transdifferentiation of HSCs to activated HSCs or myofibroblasts that produce substances of the extracellular matrix (collagen types I, III, and IV; fibronectin; laminin; and proteoglycans) [10].

3. At this point, the role of tissue hypoxia in the formation of new blood capillaries is crucial. This neoangiogenesis is the beginning of the generation of portosystemic collateral vessels and portal hypertension syndrome and can lead to the so-called “point of no return” [11,12].

In order to understand the potential treatment options for liver fibrosis, we summarize the fibrogenic effectors involved in liver fibrosis (Figure 2).

*Damaged hepatocytes*. Hepatocyte apoptosis is a strong inducer of fibrogenesis (chiefly in liver diseases with strong oxidative stress, such as ASH and NASH). Phagocytosis of apoptotic hepatocytes by myofibroblasts triggers their fibrogenic activation via NADPH oxidase 2 (NOX2) and the JAK/STAT and PI3K/Akt pathways [13].

*Activated myofibroblasts* derive from both activated hepatic stellate cells and portal fibroblasts, which are the principal producers of scar but are also involved in fibrosis regression through the release of ECM-degrading proteases [14,15]. Furthermore, several vascular mediators, like endothelins (produced by endothelial cells) and nitric oxide (NO), provoke HSC contractility. NO modulates intrahepatic resistance by stimulating a soluble guanylate cyclase that decreases Ca^2+^ levels and provoking vasodilation and HSC relaxation [16].

*Biliary progenitors*. Biliary progenitor cells (activated cholangiocytes) are involved in the attraction and activation of HSCs/myofibroblasts to proliferate and deposit ECM. They are more resistant to oxidative stress and hepatocyte death [13].

*Liver sinusoidal endothelial cells (LSECs)*. Hepatic (neo-)vascularization with LSEC activation and proliferation is tightly associated with perisinusoidal fibrosis (capillarization of the sinusoids) because LSECs contribute to ECM production and secrete both cytokines (e.g., TGF-1 and PDGF-BB), which activate HSC, and vasoconstriction factors (e.g., endothelin-1) [17].

*T cells*. Regulatory T cells appear to either favor or inhibit fibrogenesis; CD4^+^ T cells with a Th2 polarization promote fibrogenesis due to the production of IL-4 and IL-13, whereas CD4^+^ Th1 cells have an antifibrotic effect [18]. Th17 cells are clear drivers of fibrosis in multiple tissues and secrete IL-17A, which drives fibrogenesis directly in terms of myofibroblasts and indirectly via the stimulation of TGF-β1 release from inflammatory cells [19,20,21].

*Monocytes*. Monocytes are essential in inflammation and fibrosis because they are precursors of fibrocytes, macrophages, and dendritic cells. They are also involved in adaptive immune responses (proinflammatory monocytes CD14^+^CD16^+^) promoting fibrogenesis. CCL2 and its receptor CCR2 promote monocyte recruitment in the inflammatory lesion, and CXCL9 (as well as CXCL10) prevent pathological angiogenesis and fibrogenesis via the activation of their receptor, CX3CR [22,23,24].

*Macrophages*. Macrophages have a double function: they are fibrogenic during fibrosis progression and fibrolytic during its reversal. The antifibrogenic mechanism starts with the activation of Ly6C^high^ macrophages, which can differentiate into macrophages with antifibrogenic action (inactive macrophage Ly6C^low^) [6,23].

As noted, recent evidence coming from human studies and animal models has overtaken the old dogma of the irreversibility of liver fibrosis, and it is now considered a dynamic and reversible process [7]. Fibrosis reversibility starts with activated Ly6C^high^ macrophages, which can differentiate into macrophages with fibrolytic action or inactivation of macrophage Ly6C^low^ if the noxa is removed (e.g., the eradication of hepatitis C) [25]. The CX3CR1 receptor seems to be involved in such differentiation, and its overexpression is associated with the phenotypic transformation into Ly6C^low^ macrophages. These macrophages release large amounts of extracellular matrix metalloproteinases (MMP-9 and MMP-13) and the anti-inflammatory cytokine IL-10, which are involved in the resolution of fibrosis. The Ly6C^low^ macrophages lose their ability to stimulate and maintain the myofibroblast phenotype. As a result, activated stellate cells die from apoptosis (programmed cell death) or revert to quiescent cells (senescence) [26]. These concepts are summarized in Figure 2.

These data suggest that the activation of apoptosis could be a good therapeutic target of fibrosis reversibility. This approximation, however, faces the problem of the induction of fibrogenesis through apoptotic cells. Interestingly, inactivated HSCs that persist in the liver after an acute injury are much more sensitive to new transdifferentiation to myofibroblasts. Thus, in a previously damaged liver, a new injury will activate the profibrogenic transformation faster [26].

## 3. Fibrosis Reversion—Lysyl Oxidase 2

At this point, three questions arise: Can fibrosis be completely reversed into a healthy liver? Does the potential liver regeneration depend on the baseline stage of fibrosis? Is causal treatment of the liver disease enough in the early stages but not at later stages?

To date, the etiological treatment of chronic liver disease has been the basis of research in hepatology, with great success in both inhibiting the replication of hepatitis B and eradicating hepatitis C even at advanced stages of the disease with the appearance of new direct antiviral drugs. These strategies have been successful in blocking liver injury and therefore the progression to advanced fibrosis, and even achieving its reversion. In a multicenter randomized placebo-controlled trial involving 651 patients with HBV cirrhosis [27], lamivudine decreased the risk of severe liver complications (7.8% vs. 17.7%; *p* = 0.001). Similar findings have been reported in patients with advanced non-cirrhotic HCV disease, in whom the risk of liver cancer, liver failure, and all-cause and liver-related mortality decreased substantially during a ten-year follow-up in patients achieving sustained viral response (SVR) [28].

A recent study has evaluated the regression of fibrosis in 304 patients with SVR through liver stiffness measurement (LSM) using FibroScan^®^. In comparison to its baseline values, LSM was unchanged in 60%, decreased in 41%, and worsened in 7.1% of the patients. Among the 130 patients classified as F4, 78 remained stable, 22 went to F3, 11 to F2, and 19 to F0/F1 [29].

As far as the influence of viral clearance on portal hypertension is concerned, our group has recently shown that SVR was associated with a normalization of the hepatic venous pressure gradient (HVPG) in more than 50% of HCV patients with previous portal hypertension (HVPG > 6 mmHg) and with a significant reduction in the FibroScan^®^ values (<7.1 kPa) in a third of the patients [28]. The regression of hyperdynamic circulation, activation of vasoactive systems, and involution of portosystemic collateral networks are aspects currently under evaluation [30]. In a recent Spanish study with 226 cirrhotic patients (21% Child B and 75% with esophageal varices), it was demonstrated that despite the fact that the HVPG decreased from 15 mmHg to 13 mmHg at 24 weeks after treatment with direct antiviral agents (DAAs), clinically significant portal hypertension (HVPG > 10 mmHg) persisted in 78% of the cases [31].

Another issue to be taken into account in the method is the validity of elastographic (mainly Fibroscan^®^) and other serological methods in the follow-up of post-SVR fibrosis and steatosis [32,33]. Fibroscan^®^ improves after viral elimination due to the decrease in necroinflammatory activity, but we know it is not a good screening method for clinically significant portal hypertension since it was recently shown that at 24 and 96 weeks after treatment with DAAs, 43% and 28% of patients, respectively, with a Fibroscan^®^ of <13.6 Kpa have a HVPG of >10 mmHg [31].

Although patient prognosis obviously improves with viral suppression, the risk of hepatocelular hepatocarcinoma (HCC) development still exists in those patients with no regression of advanced fibrosis. The development of new antifibrogenic drugs is most needed in these patients [29,30,31,32,33,34]. In Figure 2, the potential therapeutic targets are summarized.

Recently, the research of biochemical changes affecting fibrosis irreversibility has identified lysyl oxidase-like 2 (LOXL2), an enzyme that promotes the network of collagen fibers of the extracellular matrix [35]. The lysyl oxidase gene family is currently composed of five variants (LOX and four LOX-like variants, LOXL1–4). The protein isoforms are synthesized as inactive proenzyme and secreted into the extracellular environment where they are cleaved to a mature and functional form. LOX and LOXL1–4 proteins share a highly conserved C-terminal region that contains the catalytic domain, while the N-termini are less conserved among the five members and are supposed to determine their functional role and tissue distribution [36].

LOX is secreted as proproteins (proLOX), which are proteolytically cleaved to release the free catalysts and the N-terminal propeptide regions which immediately precede the catalytic domains. ProLOX molecule is catalytically quiescent but is activated by proteolytic cleavage between Gly162 and Asp163 (rat LOX sequence) by procollagen C-proteinase. The redundancy of proteases involved in the maturation and activation of LOX underscores the importance of LOX activity in ECM homeostasis [35]. Therefore, LOX can be regulated at three levels: Synthesis of LOX precursor (hypoxia inducible factor-1 regulates the expression of the LOX gene and advanced glycation end products (AGEs) induce the binding of transcription factors); extracellular conversion of the precursor into the mature enzyme (up-regulated by humoral factors like TGF-β and down-regulated by prostaglandin E2); and direct stimulation of the activity of the enzyme (homocysteine, directly inhibit LOX activity through a direct covalent interaction with the enzyme carbonyl cofactor lysine tyrosylquinone) [36].

LOXL is a Cu-dependent amine oxidase. Copper in lysyl oxidase appears to be involved in the transfer of oxygen to facilitate the oxidative deamination of targeted peptidyl lysyl groups in tropocollagen or tropoelastin and to internally catalyse quinone cofactor formation. LOXL2 is capable of a post-transcriptional modification of type 1 collagen (the main collagen in hepatic fibrosis) and elastin, by oxidation of peptidyl lysine and hydroxylysine residue collagen, transforming it to allysine, which is responsible for the formation of crosslinks that stabilize collagen and elastin in the extracellular matrix. LOXL2 (animal and human) was detected in biliary fibrosis and NASH liver cirrhotic models [35,36,37].

Regarding oncogenesis, chronic liver disease is a well-known risk factor for HCC [28]. One of the mechanisms by which this chronic inflammation leads to fibrosis, cirrhosis, dysplastic nodules, early HCC, and in the end to advanced metastatic HCC is the induction of TGF-B, Smad4, and LOXL2. Activation of this pathway has been shown in the progression of breast cancer and the appearance of intrahepatic metastasis of HCC. The involvement of LOXL2 in the transformation of the extracellular matrix in a “prooncogenic” tissue that favors the metastatic niche of intrahepatic HCC has been recently published in *Hepatology* [38]. LOXL2 inhibition can decrease cell numbers, proliferation, colony formations, and cell growth, and it can induce cell cycle arrest and increase apoptosis. Expression levels of LOXL2 were markedly increased in matched adjacent non-tumor tissue compared with levels in tumor tissue samples, and this difference gradually increased with higher histological grade and more advanced hepatocellular tumors [39]. Hypoxia increases LOXL mRNA, LOX protein, and secreted LOXL activity, resulting in the enhanced invasive migration required for metastatic spread. While LOXL is known to be induced and/or activated by growth factors such as TGF-4, hypoxia might be more clinically relevant with regard to tumor progression [39].

While the relationship of serum LOXL2 with the stage and progression of pulmonary fibrosis has been demonstrated [40], its relationship with liver fibrosis is still under evaluation. At the Liver Meeting 2015, three poster communications did not support that changes of serum LOXL2 and changes of non-invasive markers of fibrosis or HVPG after SVR were correlated [41,42,43]. Our own group recently communicated, at the 2019 Annual Meeting of Spanish Association for the Study of the Liver, that LOXL2 serum levels decreased after viral eradication in a cohort of 271 patients with HCV liver disease two years post RVS with DAAs. However, we found a high variability of values among the patients (depending on the presence of inflammatory comorbidities, such as inflammatory diseases) and in a small cohort of pretreatment cirrhotic patients, we observed a higher tissue expression in most of them, which was related to serum expression but not to liver fibrosis stage post treatment [44].

In 2010, an interesting paper published in *Nature Medicine* [45] studied the role of LOXL2 in rodent pulmonary/liver fibrosis and cancer models and the benefit of inhibition by a monoclonal antibody (AB0023) AB0023 inhibited vessel branching, number, and length in a dose-dependent manner, with complete inhibition at the highest concentration. Treatment with monoclonal antibody (AB0023) improved mouse survival at 25 days (100% in the treated group vs. 50% in the control group, *p* < 0.006) and reversed fibrosis F3 to F1 (assessed according to the METAVIR score) in most models. Likewise, a decrease in the expression levels of αactin in myofibroblasts was observed (responsible for switching on TGF-B signaling, which activates both fibrogenesis and oncogenesis).

Given the above findings, a humanized IgG4 monoclonal antibody against LOXL2, SIMTUZUMAB^®^ (Gilead Sciences SA) has been developed [46]. Simtuzumab (GS-6624, formerly AB0024, a fully humanized variant of AB0023 with equivalent LOXL2 binding and inhibitory properties) has already been studied in idiopathic pulmonary fibrosis and colorectal and pancreatic cancer [47,48,49]. Accordingly, a recent randomized, double-blind, controlled phase 2 trial has evaluated its efficacy in more than 500 patients with idiopathic pulmonary fibrosis. The preliminary results did not support the use of Simtuzumab for patients with this disease because it did not improve progression-free survival [40,47]. Findings for colorectal and pancreatic cancer did not support a clear clinical benefit [48,49].

To date, trials in patients with liver fibrosis treated with Simtuzumab did not show improvements of clinical and fibrosis outcomes. The first report was communicated at the EASL meeting, Amsterdam, 2013. Twenty patients with liver fibrosis (F1 to F3 in METAVIR) [50] received three intravenous injections of Simtuzumab (divided in two groups with 10 mg/Kg or 20 mg/Kg, respectively) in six weeks. In both groups, an improvement in liver function tests was observed without serious side effects related to the treatment (abdominal pain, headache, muscle pain, or fatigue). The results in the regression of liver fibrosis have not been published yet. Recently, three trials in NASH and primary sclerosing cholangitis (PSC) have been published [51,52,53]. Unfortunately, Simtuzumab is ineffective in decreasing the hepatic collagen content or hepatic venous pressure gradient for patients with nonsignificant fibrosis, bridging fibrosis, or compensated cirrhosis caused by NASH in association (or not) with ASK 1 inhibitor (selonsertib) [51,52]. In PSC, Simtuzumab or placebo treatment for 96 weeks did not lead to any significant reductions in Ishak fibrosis stage, progression to cirrhosis, or frequency of clinical events, regardless of the dose (injections of 75 mg or 125 mg) [53]. The use of Simtuzumab was related with adverse events. The most common grade 3 or 4 abnormalities were elevated liver biochemistry or hyperglycemia. In the NASH patients trial a total of five patients died (two with bridging fibrosis and three with cirrhosis), three of them treated with Simtuzumab and two with placebo [51].

Despite positive preclinical data in rodent models, the failure of Simtuzumab in these three trials may be due to a number of factors: Reversibility of liver fibrosis is more pronounced in rodent models than in humans and they are probably less sensitive to liver antifibrotic therapies than rodents [34]; there is no good correlation between LOXL2 serum levels and tissue expression, and this fact could be a reason for a non-response to the treatment; third, the activity of Simtuzumab may be insufficient to inhibit collagen cross-linkage in vivo due to redundancy in other pathways that mediate collagen cross-linkage, including other LOX isoforms and tissue transglutaminase; fourth, overall in patients with cirrhosis and portal hypertension, the regression of fibrosis is less likely (point of no return); and fifth, genetic and epigenetic changes affect progression and regression of fibrosis in humans and rodents [6].

Hence, to date, the results do not support the use of monoclonal antibody against LOXL2 for the treatment of liver fibrosis. As shown in green in Figure 2, there are many potential therapies for liver fibrosis. The use of ASK 1 inhibitor (selonsertib) [52] or a pan-caspase inhibitor such as emricasan [54] could be potential therapeutic options for liver fibrosis and portal hypertension. However, the huge complexity of the fibrogenic mechanism and the fact that the mediator of liver regeneration varies between patients complicate the development of an effective drug—or, perhaps, we need combinations of drugs [12].

## 4. Non-Invasive Evaluation of Liver Fibrosis

Another key aspect is the assessment method of liver fibrosis. Liver biopsy remains the gold standard for the grading of hepatic fibrosis. However, it is not a perfect method, as it is invasive and may be subject to sampling errors. That is why the development and validation of non-invasive methods (e.g., imaging—elastographic and serological) to evaluate the regression of liver fibrosis is essential [55,56,57,58,59]. Currently we have available a lot of serum biomarkers for non-invasive evaluation of liver fibrosis in every etiology of chronic liver disease: HCV, AST to Platelet Ratio (APR)I, Forns index, Fibrotest^®^, Enhanced Liver Fibrosis score^®^ (ELF), Lok index, NASH/NALFD, NAFLD Fibrosis Score, and Fatty liver index (FLI); but none of them is validated in situations like SVR and they are very influenced by the presence of inflammatory comorbidities (rheumatoid arthritis) and the necroinflammatory activity of the liver. Therefore the interpretation of each test requires a critical analysis in order to avoid false positive or false negative results [56].

As we described above, LSM is not a good method for screening significant liver fibrosis post RVS [31]. The number of non-invasive methods has increased in recent years: Transient, vibration-controlled shear wave and magnetic resonance elastography [57,58], acoustic force radiation impulse, magnetic resonance techniques to determine the inflammation and fibrosis score [59] or the quantification of ECM molecules [60], or dynamic markers of collagen synthesis. The data suggest that these tests have a high sensitivity and specificity for the detection of advanced fibrosis and cirrhosis but are not highly sensitive or specific for less advanced stages of fibrosis. Perhaps, the approach of using quantitative liver function tests, such as cholate clearances (validated in HCV patients), metabolic tests, or SPECT liver–spleen scan, to predict the risk of future clinical outcomes is more interesting, but more studies are needed for their implementation in daily clinical practice [61,62].

## 5. Conclusions

The current therapeutic target in patients with liver cirrhosis should focus (after eradication of the causal agent) on the development of new antifibrogenic drugs. The development of these drugs must meet three premises: Patient safety, in non-cirrhotic phases, and down-staging or at least stabilization and slowing the progression to cirrhosis must be achieved; whereas in the cirrhotic stage, the objective should be to reduce fibrosis and portal pressure [7,40]. LOXL2 inhibition by Simtuzumab does not seem to be effective in liver fibrosis.

## Figures and Tables

**Figure 1 ijms-20-01634-f001:**
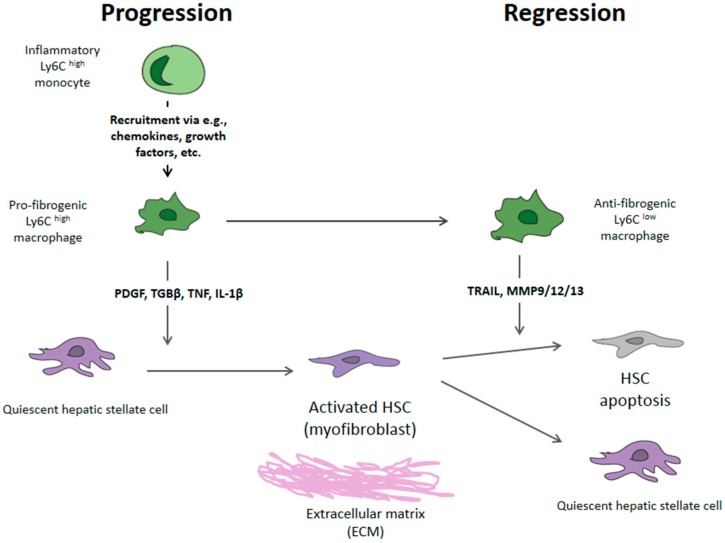
The mechanism of hepatic fibrosis: progression and regression steps. Legend: PDGF: Platelet-derived growth factor; TGBβ: Transforming growth factor β; TNF: Tumor necrosis factor; IL-1β: Interleukin, HSC: Hepatic stellate cell; TRAIL: Related apoptosis-inducing ligand receptor; MMP: Metalloproteinase; ECM: Extracellular matrix. Adapted from: Trautwein C et al., *Journal of Hepatology* 2015.

**Figure 2 ijms-20-01634-f002:**
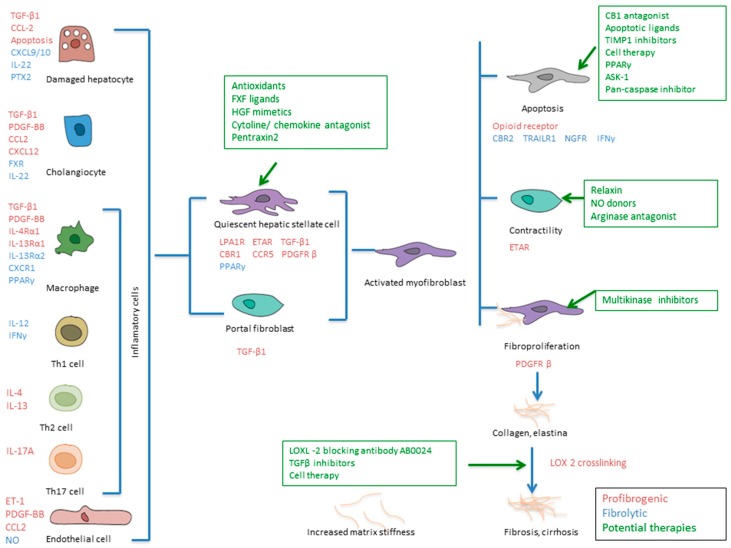
Effectors in liver fibrosis and potential therapeutic goals. Legend: Color coding: Red, profibrogenic mediators; blue, fibrolytic effectors; and green, potential therapies. Currently, there are several drugs in developement for treatment of liver fibrosis: ASK 1-inhibitor, selonsertib; pan-caspase inhibitor, emricasan; NO donors, statins; multikinase inhibitors, sorafenib, pacritinib; LOXL2, simtuzumab. Abbreviations: CXCR: Chemokine receptor; PPARγ: Peroxisome proliferator-activated receptors; CCL: Chemokine ligand; CBR: Cannabinoid receptor 1; ET-1: endothelin-1; ETAR: Endothelin A receptor; FXR: Farnesoid X receptor; Hh(R): Hedgehog (receptor); Int: Integrin; LPA1R: Lyso-phosphatidic acid receptor 1; NGFR: Nerve growth factor receptor; PTX2: Pentraxin 2; TRAIL R: Related apoptosis-inducing ligand receptor; CBR1: Carbonyl reductase 1; HGF: Hepatocyte growth factor; NGFR: Nerve growth factor receptor.

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
