# Peer review of "LOXL2—A New Target in Antifibrogenic Therapy?"

_ijms, 2019, doi:10.3390/ijms20071634_

Round 1

Reviewer 1 Report

The ms is not well prepared because a lot of typos, grammatic errors, and some of sentences lack citation.

L31 In alcoholic liver disease we found LPS binding protein or Fe; in viral hepatitis HCV/HBV proteins; in NASH, glucose, adipocytokines; PBC, bile acids and in hemochromatosis the excess iron. No citation???

L54 This inflammatory magma induces transformation of the HCS to activated HSC or

myofibroblasts that produce substances of the extracellular matrix (collagen type I, III, and IV, fibronectin, laminin and proteoglycans) [10]  transdifferentiation

L79 LSECs. Hepatic (neo-)vascularization with LSEC activation and proliferation is tightly associated with perisinusoidal fibrosis. What is LSEC???

L140 In figure 3, we summarice the potential therapeutic targets. No figure 3 is seen in this ms.

L148 LOXL2 (animal and human) was detected in HCV and NASH liver cirrhotic models. What is citation????

L157 While serum LOXL 2 and its relationship with the stage and progression of pulmonary fibrosis if have been demonstrated [34] in liver fibrosis has not been demonstrated. It is hard to read????

L162 In 2010, an interesting paper published in Nature Medicine [38, studied the role of LOXL2 in pulmonary/liver fibrosis and cancer models and the benefit of inhibition by a monoclonal antibody. What is citation???

L168 Given the above described animal models and in in vitro findings, we have developed a monoclonal antibody, SIMTUZUMAB® (Gilead Sciences SA) versus LOXL2. The Simtuzumab (GS-0 6624) is a humanized IgG4 monoclonal antibody against LOXL2 that has been already studied in pulmonary fibrosis and breast cancer. What is citation????

Author Response

Thank very much for your comments. I hardy try to improve my review, following your instructions. I attached the modified version of the manuscript. The Title of manuscript and figure 2 has been change. The english edition is certificated by IJMS.

L66 In alcoholic liver disease we found LPS binding protein or Fe; in viral hepatitis HCV/HBV proteins; in NASH, glucose, adipocytokines; PBC, bile acids and in hemochromatosis the excess iron. No citation??? 

Mediators of liver injury could be different with respect to the  etiology of the liver disease; for example, in alcoholic liver disease (ASH) we found lipopolysaccharide (LPS)-binding protein or Fe; in viral hepatitis, HCV/HBV proteins; in non alcoholic steatohepatitis, NASH , glucose and adipocytokines; in Primary biliar cholangitis,PBC , bile acids; and in hemochromatosis, excess of iron [5]. 

L81 This inflammatory magma induces transdifferentiation of the HCS to activated HSC or myofibroblasts that produce substances of the extracellular matrix (collagen type I, III, and IV, fibronectin, laminin and proteoglycans) [10]

L124 Liver Sinusoidal Endoteial Cells. Hepatic (neo-)vascularization with LSEC activation and proliferation is tightly associated with perisinusoidal fibrosis. 

L154 In figure 3, we summarice the potential therapeutic targets. No figure 3 is seen in this ms. This is a mistake. It is figure 2.

L215 LOXL2 (animal and human) was detected in HCV and NASH liver cirrhotic models. What is citation????. References [35-37]

 L230 While serum LOXL 2 and its relationship with the stage and progression of pulmonary fibrosis if have been demonstrated [34] in liver fibrosis has not been demonstrated. It is hard to read???? While the relationship of serum LOXL2 with the stage and progression of pulmonary fibrosis has been demonstrated [40], its relationship with liver fibrosis is still under evaluation

L241 In 2010, an interesting paper published in Nature Medicine [45], studied the role of LOXL2 in pulmonary/liver fibrosis and cancer models and the benefit of inhibition by a monoclonal antibody. What is citation??? Number 45.

L250 Given the above described animal models and in in vitro findings, we have developed a monoclonal antibody, SIMTUZUMAB® (Gilead Sciences SA) versus LOXL2. The Simtuzumab (GS-0 6624) is a humanized IgG4 monoclonal antibody against LOXL2 that has been already studied in pulmonary fibrosis and breast cancer. What is citation???? [46] 

Reviewer 2 Report

This is a brief review of the mechanisms of fibrosis and a more in-depth look of the role of LOXL2 in that process.  Overall, the paper is written very poorly.  The ideas are good, but the grammar and spelling make it very difficult to read and understand in most places.  It started out ok, but after the section on LOXL2 starting on Line 114, I stopped pointing out corrections to be made since they were so numerous.  Other than the grammar and spelling the authors need to address some major issues

1.  Give more detail on LOXL2.  The only thing the authors point out is that it is copper dependent and modifies collagen by oxidation of lysine and hydroxyllysine.  They need to spend more time on the mechanisms of action of the enzyme and how it is regulated.  When are copper levels low or high?  Is it activated by any of the inflammatory hormones?

2.  Since Gilead Sciences developed an antibody against LOXL2, the authors can take some time to explain how the antibody interacts with LOXL2.  This information has to be available as it has been used in clinical trials.  What were the results of this antibody in pulmonary fibrosis and breast cancer and is there any predictive action in liver?  Also, I am assuming when the authors state that “we have developed a monoclonal antibody, SIMTUZUMAB…”, they really mean people at Gilead. 

3.  In fig. 2, are the green boxes a list of activators or is that not relevant?  Why is coffee up there on the list?  It contains numerous compounds and a subset of those should be listed instead of just “coffee”

Line 24:  Scared should be scarred

Lines 31 and 32:  Confusing, rewrite.

Line 36:  Porous coating is misleading.  There is terminology for this.

Line 53:  missing word(s) at end of sentence.

Line 60:  change summarice to summarize

Lines 69-96:  There should be some mention of the effects of nitric oxide from iNOS and eNOS in response to damage to both hepatocytes and LSECs.  NO also has some effects on HSCs.

Line 78:  delete “are”

Line 80:  missing word(s)

Line 106, 113, 122-124, 125-127:  awkward sentence

Line 114:  Starting in this section, the grammar gets very poor and needs substantial improvement.  Just about every line needs some sort of grammatical or spelling improvement.

Line 133:  what is SVR?  That sentence is also awkward.

Author Response

Thank you very much for your comments. I hardy try to improve my review, following your instructions. I attached the modified version of the manuscript. The Title and figure 2 had been changed. The english edition is certificated by IJMS.

This is a brief review of the mechanisms of fibrosis and a more in-depth look of the role of LOXL2 in that process.  Overall, the paper is written very poorly.  The ideas are good, but the grammar and spelling make it very difficult to read and understand in most places.  It started out ok, but after the section on LOXL2 starting on Line 114, I stopped pointing out corrections to be made since they were so numerous.  Other than the grammar and spelling the authors need to address some major issues.

1.  Give more detail on LOXL2.  The only thing the authors point out is that it is copper dependent and modifies collagen by oxidation of lysine and hydroxyllysine.  They need to spend more time on the mechanisms of action of the enzyme and how it is regulated.  When are copper levels low or high?  Is it activated by any of the inflammatory hormones? I change and improve this part from line 197 to 215: Recently, the research of biochemical changes affecting fibrosis irreversibility has identified lysyl oxidase-like 2 (LOXL-2), an enzyme that promotes the network of collagen fibers of the extracellular matrix [35]. The lysyl oxidase gene family is currently composed of five variants (LOX and four LOX-like variants, LOXL1–4). The protein isoforms are synthesized as inactive proenzyme and secreted into the extracellular environment where they are cleaved to a mature and functional form. LOX and LOXL1–4 proteins share a highly conserved C-terminal region that contains the catalytic domain, while the N-termini are less conserved among the five members and are supposed to determine their functional role and tissue distribution [36]. LOX is secreted as proproteins (proLOX) which are proteolytically cleaved to release the free catalysts and the N-terminal propeptide regions which immediately precede the catalytic domains. ProLOX molecule is catalytically quiescent but is activated by proteolytic cleavage between Gly162 and Asp163 (rat LOX sequence) by procollagen C-proteinase. The redundancy of proteases involved in the maturation and activation of LOX underscores the importance of LOX activity in ECM homeostasis [33]. LOXL is a Cu-dependent amine oxidase capable of a post-transcriptional modification of type 1 collagen (the main collagen in hepatic fibrosis) and elastin, by oxidation of peptidyl lysine and hydroxylysine residue collagen, transforming it to allysine, responsible for the formation of crosslinks that stabilize collagen and elastin in the extracellular matrix. LOXL2 (animal and human) was detected in biliary fibrosis and NASH liver cirrhotic models [35–37].

2.  Since Gilead Sciences developed an antibody against LOXL2, the authors can take some time to explain how the antibody interacts with LOXL2.  This information has to be available as it has been used in clinical trials.  What were the results of this antibody in pulmonary fibrosis and breast cancer and is there any predictive action in liver?  Also, I am assuming when the authors state that “we have developed a monoclonal antibody, SIMTUZUMAB…”, they really mean people at Gilead.  Of course, I am an independent investigator, I was a grammar mistake, sorry for the misunderstanding.  I complete my review with the results of three recent trials in liver disease, you can see the changes between L 250 to L 273.

3.  In fig. 2, are the green boxes a list of activators or is that not relevant?  Why is coffee up there on the list?  It contains numerous compounds and a subset of those should be listed instead of just “coffee”. Whole coffee or its specific compounds appeared to decrease fatty acid synthesis (involved in steatogenesis), hepatic stellate activation (involved in fibrogenesis), and hepatic inflammation. You can check it in this recent review, Alferink LJM, Kiefte-de Jong JC, Darwish Murad S.Potential Mechanisms Underlying the Role of Coffee in Liver Health. Semin Liver Dis. 2018 Aug;38(3):193-214.

Line 51:  Scared should be scarred. Changed

Lines 53 and 65:  Confusing, rewrite.

The concept of liver fibrosis and cirrhosis being static and therefore irreversible is outdated. Indeed, both human and animal studies have shown that fibrogenesis is a dynamic and potentially reversible process that can be modulated either by stopping its progression and/or by promoting its resolution [4]. Therefore, the study of the molecular and cellular mechanisms involved in fibrogenesis is critical for establishing future antifibrotic therapies [4,5]. Furthermore, different patterns of fibrosis progression have been described on the basis of their etiology. In chronic viral hepatitis B and C, the presence of interface hepatitis and portal–central vein bridging is characteristic; in alcohol-related liver fibrosis, the deposition of ECM (extracellular matrix) in the space of Disse around sinusoids (capillarization of the sinusoids) or hepatocytes accompanied with perisinusoidal or pericellular fibrosis (also in nonalcoholic fatty liver disease) predominates. Finally, biliary fibrosis is associated with the proliferation of bile ductules and periductular myofibroblasts, which leads to the formation of portal–portal fibrotic septa [5–7].

 Mediators of liver injury could be different with respect to the etiology of the liver disease; for example, in alcoholic liver disease (ASH) we found lipopolysaccharide (LPS)-binding protein or Fe; in viral hepatitis, HCV/HBV proteins; in non alcoholic steatohepatitis, NASH, glucose and adipocytokines; in Primary biliar cholangitis,PBC, bile acids; and in hemochromatosis, excess of iron [5].

Line 74:  Porous coating is misleading.  There is terminology for this.

The liver parenchyma consists of its own epithelial cells (hepatocytes), endothelial cells, and non-parenchymal cells, including hepatic stellate cells (HSCs) and Kupffer cells (KCs) [2]. The hepatic microvascular functional unit is the liver sinusoid. It is formed of an endothelial lining distinguished by fenestrated pores and is separated from the hepatocytes by the subendothelial space of Disse, where HSCs reside [6].

Line 89:  missing word(s) at end of sentence.

Acute liver damage triggers the activation of Kupffer cells which coordinate the regenerative response; however, if chronic damage persists, over-activation of these cells results in the release of proinflammatory cytokines (CCL2 and CCL5) and a stimulation of the bone marrow for the generation of activated Ly6Chigh [8,9].

Line 99:  change summarice to summarize. Corrected

Lines 115-120:  There should be some mention of the effects of nitric oxide from iNOS and eNOS in response to damage to both hepatocytes and LSECs.  NO also has some effects on HSCs. Activated myofibroblasts derive from both activated hepatic stellate cells and portal fibroblasts, which are the principal producers of scar but are also involved in fibrosis regression through the release of ECM-degrading proteases [14,15]. Furthermore, several vascular mediators, like endothelins (produced by endothelial cells) and nitric oxide (NO), provoke HSC contractility. NO modulates intrahepatic resistance by stimulating a soluble guanylate cyclase that decreases Ca2+ levels and provoking vasodilation and HSC relaxation [16].

Line 78:  delete “are” Corrected

Line 80:  missing word(s).  Line 106, 113, 122-124, 125-127:  awkward sentence Line 114:  Starting in this section, the grammar gets very poor and needs substantial improvement.  Just about every line needs some sort of grammatical or spelling improvement. Line 133:  what is SVR?  Sustained virologycal response . That sentence is also awkward.

I change a lot of sentences and the grammar has been review and improved as you can observe in the manuscriptV1.

Round 2

Reviewer 1 Report

In abstract, Lysyl oxidase 2 should be included.

Any evidence shows that non-invasive examination is correlated with biomarker or biochemical analysis in liver fibrosis.

Too many abbrs, suggest the authors can give an abbreviations list.

15 mmHg to 13 mmHg need more space between value and unit.

References 36,41,43, the format is correct or not, double check.

Author Response

Thank you very much for your comments. I hardy try to improve my review, following your instructions. I attached the modified version of the manuscript and figure 2.

In abstract, Lysyl oxidase 2 should be included. The research of biochemical changes affecting fibrosis irreversibility has identified lysyl oxidase-like 2 (LOXL-2), an enzyme that promotes the network of collagen fibers of the extracellular matrix. LOXL2 inhibition can decrease cell numbers, proliferation, colony formations, and cell growth, and it can induce cell cycle arrest and increase apoptosis. And  tThe development of a new a humanized IgG4 monoclonal antibody against LOXL2, could open the window of a new antifibrogenic treatment.

Any evidence shows that non-invasive examination is correlated with biomarker or biochemical analysis in liver fibrosis. Currently we have available a lot of serum biomarkers for non-invasive evaluation of liver fibrosis in every etiology of chronic liver disease: HCV , AST to Platelet Ratio (APR)I, Forns index;  Fibrotest®,Enhanced Liver Fibrosis score® (ELF), Lok index; NASH/NALFD, NAFLD Fibrosis Score, Fatty liver index (FLI); but none of them is validated in situations like SVR and they are very influenced by the presence of inflammatory comorbidities (rheumatoid arthritis) and the necroinflammatory activity of the liver. Therefore the interpretation of each test requires a critical analysis in order to avoid false positive or false negative results.

Too many abbrs, suggest the authors can give an abbreviations list. I complete de abbreviations list with all the abbrs used. HSC, hepatic stellate cell; HCV, hepatitis C virus; HBV, hepatitis B virus; NASH, nonalcoholic steatohepatitis; ASH, alcoholic steatohepatitis; PBC, primary biliary cirrhosis; KC, Kupffer cells; EM, extracellular matrix; EMC, extracellular matrix component; CXCR, chemokine receptor; CX3CR1, fractalkine receptor; PPARγ, peroxisome proliferator-activated receptors; CCL, chemokine ligand; MMP, metalloproteinases; SVR, sustained virological response, . LOXL-2, lysyl oxidase-like 2; LSECs ,Liver sinusoidal endothelial cells; LSM, liver stiffness measurement; HCC hepatocelular hepatocarcinoma; DAAs, direct antiviral agents; NO nitric oxide; PDGF, platelet-derived growth factor; TGBβ, transforming growth factor β; TNF, tumor necrosis factor; IL-1β, interleukin, TRAIL, related apoptosis-inducing ligand receptor; CBR, cannabinoid receptor 1; ET-1, endothelin-1; ETAR, endothelin A receptor; FXR, farnesoid X receptor; Hh(R), hedgehog (receptor); Int, integrin; LPA1R, lyso-phosphatidic acid receptor 1; NGFR, nerve growth factor receptor; PTX2, pentraxin 2; CBR1, carbonyl reductase 1; HGF, hepatocyte growth factor; NGFR, nerve growth factor receptor.

15 mmHg to 13 mmHg need more space between value and unit. Line 189 change: “demonstrated that despite the fact that the HVPG decreased from 15 mmHg to 13 mmHg”

References 36,41,43, the format is correct or not, double check.

36. Lucero H  and H. M. KaganH Lysyl oxidase: an oxidative enzyme and effector of cell function. Cell. Mol. Life Sci. 2006 (63): 2304–2316

41. Bosch et al. Correlation between noninvasive markers of fibrosis and the hepatic venous pressure gradient (HVPG) in patients with compensated cirrhosis due to nonalcoholic steatohepatitis (NASH). Hepatology 2015, Oct 2015;62 (S1), 120A

42. Afdha et al Serum lysyl oxidase-like-2 (sLOXL2) is correlated with the hepatic venous pressure gradient (HVPG) in patients with cirrhosis due to hepatitis C. Hepatology 2015, Oct 2015;62 (S1), 121A

43. Bourliere et al. Changes in liver stiffness by transient elastography (TE) and serum lysyl oxidase-like-2 (sLOXL2) in patients with cirrhosis treated with ledipasvir/sofosbuvir (LDV/ SOF)-based therapy. Hepatology 20155, Oct 2015;62 (S1), 123A

Reviewer 2 Report

Revision of IJMS-440651

The spelling and grammar of the revised manuscript is appreciated.  Some of my previous concerns were not addressed as the authors lightly touched upon issues that require a deeper analysis for the conclusions that are made for this work.

1.  The text between lines 268 and 284 need more development.  Yes, I can go and read the referenced papers, however, a few lines detailing those findings are in order so that the reader of this manuscript will not be left with more questions than answers.  How is AB0023 different from SIMTUZUMAB (GS-6624)?  Is AB0024 different than GS-6624?  Are human and rodent LOXL2 enzymes significantly different from each other that they have different activities?  Some basic biochemical explanation is in order here to understand why one antibody seems to work in rodents and not in humans.

2.  In fig. 2, are the green boxes a list of activators or is that not relevant?  Why is coffee up there on the list?  It contains numerous compounds and a subset of those should be listed instead of just “coffee”.  The authors should eliminate coffee if the chemical source is not identified.  The figure legend should define the color coding.  I am not sure why there are 3 of these figures.

3.  Give more detail on LOXL2.  The only thing the authors point out is that it is copper dependent and modifies collagen by oxidation of lysine and hydroxyllysine.  They need to spend more time on the mechanisms of action of the enzyme and how it is regulated.  When are copper levels low or high?  Is it activated by any of the inflammatory hormones?

Minor comments

Line 320:  goof should be good

Author Response

Thank you very much for your comments. I hardy try to improve my review, following your instructions. I attached the modified version of the manuscript and figure 2.

The spelling and grammar of the revised manuscript is appreciated.  Some of my previous concerns were not addressed as the authors lightly touched upon issues that require a deeper analysis for the conclusions that are made for this work.

1.  The text between lines 268 and 284 need more development.  Yes, I can go and read the referenced papers, however, a few lines detailing those findings are in order so that the reader of this manuscript will not be left with more questions than answers.  How is AB0023 different from SIMTUZUMAB (GS-6624)?  Is AB0024 different than GS-6624?  Are human and rodent LOXL2 enzymes significantly different from each other that they have different activities?  Some basic biochemical explanation is in order here to understand why one antibody seems to work in rodents and not in humans.

Given the above findings, a humanized IgG4 monoclonal antibody against LOXL2, SIMTUZUMAB® (Gilead Sciences SA) has been developed [46]. Simtuzumab (GS-6624,formerly AB0024,a fully humanized variant of AB0023 with equivalent LOXL2 binding and inhibitory properties).

Despite positive preclinical data in rodent models,he failure of Simtuzumab in these three trials may be due to a number of factors: reversibility of liver fibrosis is more pronounced in rodent models than in humans [55]; there is no good correlation between LOXL-2 serum levels and tissue expression, and this fact could be a reason for a non-response to the treatment. SecondThird, the activity of Simtuzumab may be insufficient to inhibit collagen cross-linkage in vivo , and third, LOXL2 inhibition may be ineffective due to redundancy in other pathways that mediate collagen cross-linkage,including .other LOX isoforms and tissue transglutaminase; fourth, overall in patients with cirrhosis and portal hypertension  the regression of fibrosis is less likely (point of no return) and fifth genetic and epigenetic changes affects progression and regression of fibrosis in humans and rodents [6].

2.  In fig. 2, are the green boxes a list of activators or is that not relevant?  Why is coffee up there on the list?  It contains numerous compounds and a subset of those should be listed instead of just “coffee”.  The authors should eliminate coffee if the chemical source is not identified.  The figure legend should define the color coding.  I am not sure why there are 3 of these figures.

I remove coffee following your instructions and the color coding in the legend. Also I add, the drugs currently in study for liver fibrosis.

Effectors in liver fibrosis and potential therapeutic goals. Legend: Color coding :red profibrogenic mediators; blue; fibrolytic effectors and green potential therapies. Currently,  there are the several drugs in developement for treatment of liver fibrosis: ASK 1-inhibitor, selonsertib; pan-caspase inhibitor,emricasan; NO donors, statins; multikinase inhibitors, sorafenib, pacritinib; LOXL-2, simtuzumab. Abbreviations: CXCR: chemokine receptor; PPARγ: peroxisome proliferator-activated receptors; CCL: Chemokine ligand; CBR: cannabinoid receptor 1; ET-1: endothelin-1; ETAR: endothelin A receptor; FXR: farnesoid X receptor; Hh(R): hedgehog (receptor); Int: integrin; LPA1R: lyso-phosphatidic acid receptor 1; NGFR: nerve growth factor receptor; PTX2: pentraxin 2; TRAIL R: elated apoptosis-inducing ligand receptor; CBR1: carbonyl reductase 1; HGF: hepatocyte growth factor; NGFR: nerve growth factor receptor.

3.  Give more detail on LOXL2.  The only thing the authors point out is that it is copper dependent and modifies collagen by oxidation of lysine and hydroxyllysine.  They need to spend more time on the mechanisms of action of the enzyme and how it is regulated.  When are copper levels low or high?  Is it activated by any of the inflammatory hormones?

LOX is secreted as proproteins (proLOX) which are proteolytically cleaved to release the free catalysts and the N-terminal propeptide regions which immediately precede the catalytic domains. ProLOX molecule is catalytically quiescent but is activated by proteolytic cleavage between Gly162 and Asp163 (rat LOX sequence) by procollagen C-proteinase. The redundancy of proteases involved in the maturation and activation of LOX underscores the importance of LOX activity in ECM homeostasis [35]. Therefore, LOX can be regulated at three levels: synthesis of LOX precursor (hypoxia inducible factor-1 regulate the expression of the LOX gene and advanced glycation end products (AGEs) induce the binding of transcription factors); extracellular conversion of the precursor into the mature enzyme ( up regulated by humoral factors like TGF-β and down regulated by prostaglandin E2); and and direct stimulation of the activity of the enzyme (homocysteine,  directly inhibit LOX activity through a direct covalent interaction with the enzyme carbonyl cofactor lysine tyrosylquinone)[36]

LOXL is a Cu-dependent amine oxidase .Copper in lysyl oxidase appears to be involved in the transfer of oxygen to facilitate the oxidative deamination of targeted peptidyl lysyl groups in tropocollagen or tropoelastin and to internally catalyse quinone cofactor formation. LOXL-2 is  capable of a post-transcriptional modification of type 1 collagen (the main collagen in hepatic fibrosis) and elastin, by oxidation of peptidyl lysine and hydroxylysine residue collagen, transforming it to allysine, responsible for the formation of crosslinks that stabilize collagen and elastin in the extracellular matrix. LOXL2 (animal and human) was detected in biliary fibrosis and NASH liver cirrhotic models [35–37].

Minor comments

Line 320:  goof should be good

Line 331: As we described above, LSM is not a good method for screnning significant liver fibrosis post RVS.

Round 3

Reviewer 2 Report

Other than some minor spelling/grammar mistakes, it should be ready for publication.